# Development of a Broad-Spectrum Antiserum against Cobra Venoms Using Recombinant Three-Finger Toxins

**DOI:** 10.3390/toxins13080556

**Published:** 2021-08-10

**Authors:** Bing-Sin Liu, Bo-Rong Jiang, Kai-Chieh Hu, Chien-Hsin Liu, Wen-Chin Hsieh, Min-Han Lin, Wang-Chou Sung

**Affiliations:** 1National Institute of Infectious Diseases and Vaccinology, National Health Research Institutes, Miaoli 35053, Taiwan; bingsin@nhri.edu.tw (B.-S.L.); office0928672666@gmail.com (B.-R.J.); apple11763@nhri.edu.tw (K.-C.H.); hamlyn@nhri.edu.tw (M.-H.L.); 2Centers for Disease Control, Ministry of Health and Welfare, Taipei 10050, Taiwan; liuch@cdc.gov.tw (C.-H.L.); vac@cdc.gov.tw (W.-C.H.)

**Keywords:** recombinant protein, three finger toxins, immunization, cobra, antiserum, broad spectrum

## Abstract

Three-finger toxins (3FTXs) are the most clinically relevant components in cobra (genus *Naja*) venoms. Administration of the antivenom is the recommended treatment for the snakebite envenomings, while the efficacy to cross-neutralize the different cobra species is typically limited, which is presumably due to intra-specific variation of the 3FTXs composition in cobra venoms. Targeting the clinically relevant venom components has been considered as an important factor for novel antivenom design. Here, we used the recombinant type of long-chain α-neurotoxins (P01391), short-chain α-neurotoxins (P60770), and cardiotoxin A3 (P60301) to generate a new immunogen formulation and investigated the potency of the resulting antiserum against the venom lethality of three medially important cobras in Asia, including the Thai monocled cobra (*Naja kaouthia*), the Taiwan cobra (*Naja atra*), and the Thai spitting cobra (*Naja Siamensis*) snake species. With the fusion of protein disulfide isomerase and the low-temperature settings, the correct disulfide bonds were built on these recombinant 3FTXs (r3FTXs), which were confirmed by the circular dichroism spectra and tandem mass spectrometry. Immunization with r3FTX was able to induce the specific antibody response to the native 3FTXs in cobra venoms. Furthermore, the horse and rabbit antiserum raised by the r3FTX mixture is able to neutralize the venom lethality of the selected three medically important cobras. Thus, the study demonstrated that the r3FTXs are potential immunogens in the development of novel antivenom with broad neutralization activity for the therapeutic treatment of victims involving cobra snakes in countries.

## 1. Introduction

Snake envenoming remains a threat to public health worldwide and leads to an estimated 1,841,000 snakebite cases and over 94,000 fatalities per year [1,2]. In Asia, approximately 12–34% of snakebite envenoming was estimated to be caused by cobra species [3,4,5,6]. Administration of the antivenom is the recommended approach for the therapy of snakebite envenomation, which consists of neutralizing antibodies to block the toxicity of venom components. According to the hospital reports and records, there are nine cobra species ranked as category 1 in Asia regions—i.e., the most medical important species responsible for the highest morbidity and mortality [7]. Only a few species-specific antivenoms are available (e.g., antivenoms for *Naja kaouthia*, *Naja atra*, *Naja naja, Naja philippinensis*, and *Naja sputatrix*) [8,9,10,11,12], while their neutralization efficacy is typically limited to the species with the same or similar toxin composition [13]. In 2017, the WHO restated that snakebite is a neglected tropical disease and aimed to develop effective strategies to decrease the life-threatening accidents by 50% before 2030 [14,15,16].

Clinically, cobra envenoming could cause severe respiratory paralysis or death of the victims, which are likely linked to major component of 3FTXs in cobra venom. 3FTXs are non-enzymatic proteins that are frequently found in the Elapid snake family. The members of this group contain different toxic functions that can be generally divided into subtypes, including the cytotoxic cardiotoxins (CTX) that can cause local tissue necrosis, and the neurotoxic alpha-neurotoxins (αNTXs) that can bind the muscle nicotinic acetylcholine receptor to cause the neuromuscular paralysis [17]. Based on the sequence length and number of the disulfide bonds, the αNTXs are subdivided into type 1-αNTX (short-chain neurotoxin, sNTX) and type 2-αNTX (long-chain neurotoxin, LNTX) [18]. As the principal elements contribute to the venom toxicity, 3FTXs have been recognized as the main targets for the neutralization antibodies. However, the conventional immunization with crude venom was not able to effectively induce specific antibodies in animal antiserum against 3FTX proteins, which is majorly due to their weak immunogenicity [19]. Additionally, 3FTX subtypes possess the toxicity sites located at various segments of the molecule surface, and antibodies targeting the antigenic epitopes at these toxicity sites play an important role for toxin neutralization [20]. As the distinct epitopes are shown on 3FTX of different subtypes, the venom antigenicity might vary with the 3FTXs composition, which limits the potency of the resulting antiserum to the cobra venom with similar toxin composition [21].

Considering that the neutralization antibodies are directed majorly against the 3FTX components, immunization with the snake venoms mixture [22] or the extracted principal toxins mixture [23] has been shown to expand the neutralization scope of the resulting antiserum. However, the availability of geographic cobra venoms or purified toxins can be a challenge task in the production of this antivenom. To overcome the obstacles, using the recombinant antigens can be an attractive alternative in the preparation of novel antivenoms. To date, series modern materials, such as peptides [24,25], DNA (or virus vector) [26], or recombinant proteins [27,28,29], have been developed and showed the potency to generate the antiserum with neutralization potency. Of them, the full-length recombinant proteins showed the capability to induce a wide range of antibodies recognizing the potential domains covering the whole sequence or targeting specific conformation sites of toxin. Previously, Jaime et al. used the *E. coli* to express the consensus short-chain αNTXs and demonstrated they were immunogenic to induce the protective antibodies against the lethality of native toxins [27] and sNTX-dominant venoms [28,29]. Such results emphasized the potential use of recombinant proteins instead of venoms to generate effective antiserum.

In the study, three principal 3FTXs, LNTX (P01391), sNTX (P60770), and CTXA3 (P60301) were selected based on their abundance and lethality [30]; an *E. coli* heterologous expression system was used to express these recombinant toxin candidates to assess their potential applications for the development of novel antivenoms. As 3FTXs are cysteine-rich proteins, and the correct disulfide bond pattern is essential for the folding of the native conformation and subsequent effective induction of antibodies [31], a thiol-disulfide isomerase Disulfide bond C (DsbC) was fused with the 3FTX candidate during the bacterial expression to assist with correct disulfide bond rearrangement; DsbC has been extensively used for the expression of recombinant proteins with multiple disulfide bonds [32,33,34]. Finally, the immunogenicity of r3FTXs was thoroughly analyzed in mice, rabbits, and horse, and the neutralizing potency of the resulting antiserum against the lethality of venoms of *N. atra*, *N. kaouthia*, and *N. siamensis* was evaluated.

## 2. Results

### 2.1. Expression of Recombinant Three-Finger Toxins (r3FTXs)

Three 3FTX candidates of different subtypes were expressed with the *E. coli* system. To construct the correct disulfide bond linkages in the recombinant proteins, each r3FTX gene was fused with the DsbC gene for heterologous expression in *E. coli*, as shown in Figure 1A. After IPTG induction, an abundance of His-tagged fusion proteins were induced in the cytoplasm of *E. coli*, and the proteins were then directly purified by immobilized metal affinity chromatography (IMAC) affinity chromatography. Tobacco etch virus (TEV) protease digestion and sequential purification using ion-exchange columns yielded the recombinant type of sNTX (rsNTX), LNTX (rLNTX) and CTX A3 (rCTXA3) at the milligram level. The purity of each r3FTX was confirmed by the presence of a single band, as shown in SDS-PAGE (Figure 1B,D,F). The levels of lipopolysaccharide (LPS) in the final products were below 10 EU/mg. Mass spectrometry analysis on the purified r3FTXs detected a mass difference of 57 Da as compared to that of their corresponding native toxins, which is a glycine residue (glycine, C2H3NO) derived from the TEV linkage peptide after protease digestion (Table 1). Furthermore, the Western blotting indicated that all r3FTXs were recognized by the corresponding antivenom antibodies (Figure 1C,E,G), suggesting that each of the recombinant toxins shared similar linear antigenic epitopes with the corresponding native ones.

### 2.2. Structural Characterization of r3FTXs

At the beginning of the study, circular dichroism (CD) spectrometry was applied to characterize the secondary structure of r3FTXs to compare the structural conformation with that of the native ones. The CD spectra of r3FTXs had a positive peak at 199 nm and a negative peak at 211 nm that are typical for the β-sheet secondary structure (Appendix A), and in silica analsyis on the acquired spectra revealed that these r3FTXs had a high content of β-sheet conformation similar to the native toxins [35], as shown in Table 1. Additionally, a low percentage of α-helices was observed on the LNTX and rLNTX, which results from a cyclic loop formed at the tip of loop II by a disulfide bridge [36]. The formation of native like secondary structures is highly correlated with matched disulfide bond patterns formed in r3FTXs. Through the tandem mass spectrometry (MS/MS) characterization, the peptides with matched disulfide linkages were identified from the tryptic digest of r3FTXs, which presented the similar collision-induced dissociation (CID) fragments (a1, b, and y ions) as that of disulfide-linked peptides derived from the native ones (Appendix A).

In addition to the physical characterization, the lethal dosage of r3FTX was determined with the animal assay. As shown in Table 1, the recombinant α-neurotoxins, rLNTX and rsNTX, present the similar lethality as that of the native ones, which can cause the mice to die within hours, possibly due to the neurotoxic effect. In contrast, the toxicity of CTX has been reported to lysis cell membranes, which is likely linked to the severe tissue swelling syndromes in cobra-bitten victims [37]. Through the in vitro cell-based assay, rCTXA3 presented a comparable cytotoxic effect on the HL-60 cells (IC_50_ 1.93) as that of the native CTXA3 (IC_50_ 2.02). Denaturing r3FTXs by a reducing agent, i.e., DTT, resulted in the loss of all toxicity, which is in agreement with our previous findings in native CTX A3 toxin [31]. Overall, the physical characterizations and toxicity assay presented the similarity on the structural properties of the expressed r3FTXs and the corresponding 3FTXs, which highlighted the potency of the *E coli* system in preparing the native-like recombinant toxins.

### 2.3. Assessment of the Immunogenicity of Individual r3FTX in a Mouse Model

In the study, groups of 5 mice were intramuscularly immunized three times at 2-week intervals with 50 μg of antigen (i.e., rsNTX, rLNTX, or rCTX A3) to assess the immunogenicity for antibody induction. The mouse sera were collected two weeks after the final immunization [29,38], and the ELISA assay results demonstrated that these antisera can recognize the corresponding toxin purified from the crude venoms (Figure 2A–C). Meanwhile, the pre-immune mice sera did not show the reactivity to the coating antigens in the assay. The geometric mean titration of mouse sera raised by either rsNTX or rLNTX could reach 10^5^, which is comparable to the titers of mice sera raised against the native ones (Figure 2A,B). Meanwhile, mouse sera raised by rCTXA3 presented a lower titration against the native CTX A3 as compared to that of mouse sera raised by native CTX A3 (Figure 2C), which is possibly due to the variance of breeding mice or a trace amount of containment in the purified CTX A3 immunogens contributing to the higher antibody titers in the assay [39].

### 2.4. Assessment of Immunogenicity of r3FTX Mixture in Rabbit Model

The immunization of rabbits is a common step prior to testing novel immunogens in horses. In the study, a rabbit was immunized five times at a 2-week interval with the mixture of three r3FTXs to assess the potency in the induction of protective antibodies. A mixture of equal amounts of *N. atra* and *N. kaouthia* venoms was prepared to immunize another rabbit for the comparative analysis. Notably, the composition of three r3FTXs mixture was manipulated to mimic the abundance of the corresponding toxins in the venom mixture. After a prime injection and four rounds of boost injections at an interval of 2 weeks, comparable titers (>10^5^) of toxin-specific antibodies were detected in rabbit antisera raised by either immunogen, as shown in Table 2.

### 2.5. Immunoreactivity and Neutralizing Potency of Rabbit Anti-r3FTXs Antibodies

Accordingly, an antivenomic approach was conducted to assess the cross-reactivity of the rabbit anti-r3FTXs antibodies toward the venom components. By injecting the cobra venom (100 μg) into the affinity column containing the immobilized rabbit anti-r3FTXs antibodies, the flow through and elution fractions were collected and then analyzed with the reverse-phase high-performance liquid chromatography (RP-HPLC). Figure 3A,D presents the profile of *N. atra* and *N. kaouthia* venom proteins separated by RP-HPLC. By comparison, the RP-HPLC chromatograms reveled the majority in the elution fractions were αNTXs and CTXs (Figure 3B,E). In contrast, the flow-through fractions comprised the PLA2 and high molecular venom proteins, as shown in Figure 3C,F. These antivenomic results revealed that antibodies raised by the mixture of r3FTXs possessed the specificity to target the 3FTXs in the complex mixture of crude venoms and presented the intense affinity to trap the 3FTX proteins in column. Additionally, the elution chromatograms also revealed that 3FTX homologous could be trapped in the column. Such cross-reactivity might result from the high sequence identity between the r3FTXs with other 3FTX homologous [40,41,42,43,44,45,46]. For example, the sequence of rCTXA3 shared more than 80% identity with other CTX analogous in the venoms of cobra snake species. It is likely that the recombinant αNTXs shared more than 70–80% identity with other αNTXs analogous in the venoms of Asiatic *cobra* snakes (Appendix A). Such high sequence identity suggested that each r3FTX and its homologous might possess the similar antigenic sites for antibody recognition.

Furthermore, the serum antibodies were purified from the rabbit antiserum to assess the neutralization efficacy against the lethality of cobra venom, including *N. atra* (i.p. LD_50_ 0.67)*, N. kaouthia* (i.p. LD_50_ 0.15)*,* or *N. siamensis* venoms (i.p. LD_50_ 0.19). Individual cobra venom (3 × LD_50_) was pre-incubated with the serial diluted rabbit serum antibodies, and then, the mixture was injected into the mice (*n* = 6/group) to determine the potency based on the survival rate of mice within 48 h. Results from the neutralization assay revealed that rabbit serum antibodies raised by the r3FTXs were able to neutralize the lethality of the selected cobra venoms, and the potency is comparable to the rabbit serum antibodies raised by a mixture of cobra venom, as shown in Table 3. These experimental findings demonstrated that r3FTXs mixture immunization were able to induce the specific antibody response in rabbit, which were able to target native 3FTXs and neutralize the lethality of crude venoms.

### 2.6. Hyperimmunization with r3FTXs in Horse Model and Potency Evaluation of Horse Antiserum

Base on the results above, the r3FTXs hyperimmunization was performed in horse to assess the efficacy for inducing effective antibodies. By subcutaneously injecting r3FTXs mixture formulated in Freund’s adjuvant at a two-week interval, the specific antibody response was detected in horse serum at the stage after the second immunization (Week 6). Accordingly, ELISA assay revealed that the antibody titrations increased with the number of immunizations and reached the peak at the stage after the sixth immunization (week 12), as shown in Figure 4. An addition of boosting immunization was not able to stimulate higher antibody titers (Week 14). Subsequently, the horse serum collected at week 16 were analyzed for neutralization titers. Results from the neutralization assay revealed that the horse antiserum raised by r3FTXs could effectively cross-neutralize the lethal dosage of *N. atra* (3 × LD_50_), *N. kaouthia* (5 × LD_50_), and *N. siamensis* (3 × LD_50_) with the potency of 21.33, 6.10, and 25.78 mg/g, respectively (Table 3). In contrast, pre-immune horse serum was not able to neutralize the lethality of cobra venom in the assay. Consistent with the findings in the rabbit immunization experiment, the hyperimmunization with r3FTXs was able to induce the humoral response in horse animal, and the acquired serum antibodies presented the potency to neutralize the venom lethality of three medically important cobras.

## 3. Discussion

Cobra envenoming remains a threat to public health in Asia. The antivenom is the validated therapeutics for the emergency snakebite treatment; however, the cross-species neutralization capability of currently available commercial antivenom products is generally effective to the venom in a similar manner to the toxin composition as immunogen. As a variety of 3FTXs contribute to the different toxicity of cobra venoms, the new design of the immunization protocol has been considered to broaden the potency of the antivenom products. In this study, we evaluated the potential of recombinant toxins in the development of animal antiserum against three medically important cobra snakes in Asia. For cysteine-rich 3FTXs, disulfide bonds linkage plays an important role in the construction of correct protein conformation, which is critical for effective antibody induction. With the fusion of DsbC isomerase, three r3FTXs with correct disulfide bonds were generated with the *E coli* expression systems, which was confirmed by MS and CD characterization. Regarding the immunogenicity of *E coli*-expressed toxin, animal experiment results showed that each r3FTX was immunogenic to induce the specific antibodies against the native toxin in cobra venoms (Figure 1B–D), which reflected that their antigenicity was similar with the native one. As different subtypes of r3FTXs with native-like structure were generated by *E coli* in this study, it demonstrated that the platform is applicable in expressing the disulfide-rich proteins.

Generally, cobra venom contains various types of 3FTXs that could coordinate to cause the severe syndromes or lethal effect on the envenomed victims. Indeed, antibodies targeting the 3FTXs of different subtypes play an important role in neutralizing the venom toxicity. In the study, immunization with an r3FTXs mixture was shown to induce high humoral response in rabbit. Based on the findings in antivenomic study, the rabbit serum antibodies raised by the r3FTXs mixture presented a high affinity to the corresponding 3FTXs and their homologous components in cobra venoms, which can trap them from the complex venom components in the affinity column. Furthermore, neutralization assay demonstrated the potency of the rabbit anti-r3FTXs antibodies to neutralize the lethality of three medically important cobra venoms. These data showed that the immunization with r3FTXs mixture was able to induce antibodies to recognize the native 3FTX of different subtypes in cobra venoms, which might correlate to the cross-neutralization potency of the resulting rabbit antiserum.

Horse animals have been commonly used to produce antivenom in countries. To date, the serum antibodies purified from the plasma of venom-hyperimmunized horse is applied as the major therapeutics for clinical treatment of the envenomed victims. In the study, hyperimmunization with r3FTXs was performed in horse. ELISA assay showed that the antibody titers were increasing with the immunization numbers and reach a plateau at the 12th week after six rounds of immunization. In the neutralization assay, the purified horse serum antibodies showed the potency to neutralize the lethality from venom of *N. atra, N. kaouthia*, and *N. siamensis*, which is consistent with the findings observed in rabbit antiserum. These experimental findings demonstrated the potency of r3FTXs for inducing the neutralization antibodies in animals and also provide evidence that the use of principal toxins as immunogen can be an effective strategy to develop antivenom with broad potency.

## 4. Conclusions

In summary, the study presented that r3FTXs are effective immunogens in generating the animal antiserum with cross-neutralization potency against cobra venoms with different 3FTX compositions. In contrast to native toxins, the production of the recombinant toxins can be scaled up by means of bioreactors and an optimized fermentation process, and it provides a constant and cost-effective source of immunogens to improve the antivenom productivity. Considering the intra-species variations in cobra venoms from various regions of Asia, the current study demonstrated a rational and flexible design of recombinant immunogen, which can be used to replace native snake venoms to an extent to generate a multivalent cobra antivenom. In the future, more types of recombinant principal toxins of Elapid snake species may be generated to expand the neutralization scope of current antivenoms; this venom-independent strategy is expected to be useful in the development of better antivenom products to ultimately benefit the snakebite management in area without access to commercial antivenoms.

## 5. Materials and Methods

### 5.1. Venoms and Antivenoms

The venoms of *N. kaouthia* and *N. siamensis* originating from Thailand were purchased from Latoxan (Valence, France). *N. atra* venom was acquired from a local snake farm in Taiwan. The venom was lyophilized and stored at −20 °C until further use. A freeze-dried neurotoxic antivenom (FNAV, batch 61-06-0002, expiration date 23 December 2019) against *Bungarus multicinctus* and *N. atra* was acquired from Taiwan Centers for Disease Control (Taiwan CDC). The neuro polyvalent antivenom (NPAV, Lot: NP00115, expiry date: 24 March 2020) produced by Queen Saobhava Memorial Institute (QSMI, Bangkok, Thailand) was a kind gift from Dr. Dong-Zong Hung (Division of Toxicology, China Medical University Hospital, Taichung, Taiwan). Both antivenoms were used for Western blotting. NPAV is a polyspecific antivenom prepared against the venoms of Malayan Krait (*B. candidus*), Cobra (*N. kaouthia*), Banded Krait (*B. fasciatus*), and King cobra (*O. hannah*). Both antivenom immunoglobulins were in the form of lyophilized F(ab’)2 purified from the equine antisera. The antivenoms were reconstituted with deionized water for assay applications. The antivenom products and the reconstituted solution were stored at 4 °C until further use.

### 5.2. Expression of r3FTXs

The expression of r3FTXs was performed according to the procedure developed by Nozach et al. [34] with some modifications. The codon-optimized gene sequence of DsbC (651 base pairs, bp, accession no.: CAQ33205) and gene sequences of cobra αNTXs (186 bp for cobrotoxin, accession no.: P60770; 213 bp for alpha-elapitoxin-Nk2a, accession no.: P01391) or CTXA3 (180 bp, accession no.: P60301) were synthesized and linked together by a 39-nucleotide linker (5′-catcatcaccaccaccacgagaacctgtactttcagggc-3′), which encodes a His-tag (6 × histidine) followed by a TEV protease peptide (ENLYFQG). Then, the resulting gene was digested with the NdeI and BamHI restriction enzymes and subcloned into the pET-9a plasmid under the control of the T7 promoter. The resulting plasmids, named pDsbC-His-sNTX, pDsbC-His-LNTX, and pDsbC-His-CTXA3, were separately transformed into either BL21(DE3) or Rosetta (DE3) cells for heterologous expression, which was performed at 37 °C with shaking at 4 G-force in a 2 L standard flask containing 800 mL of LB growth medium with 34 mg/L kanamycin. The expression of DsbC-sNTX and DsbC-LNTX were induced at 37 °C in bacteria that reached the exponential growth phase by adding 1 mM isopropyl β-D-1-thiogalactopyranoside (IPTG), and the cells were harvested three hours after the induction; for the expression of DsbC-CTXA3, the induction temperature was set at 12 °C, and the cells were harvested the next day. Cells were disrupted by a French press, and r3FTXs were isolated from the soluble fraction of the cell lysate by an IMAC column. The eluent was dialyzed against phosphate-buffered saline (PBS) and cleaved by TEV protease at a ratio of 1:3 (TEV: Protein) in a solution containing 0.1 mM dithiothreitol (DTT) at 16 °C overnight. Subsequently, the mixture was dialyzed against sodium phosphate buffer (20 mM, pH = 5.8) and loaded onto an IMAC column to remove TEV protease, DsbC, and the parental chimera protein, which carried a hexahistidine tag. The flow-through from the IMAC column was collected and loaded onto a sulfopropyl strong cation exchange chromatography resin (GE Healthcare) to concentrate r3FTXs. Q Sepharose fast flow resin (GE Healthcare) was used to remove endotoxins at the last step. The level of endotoxin in the final r3FTXs was determined to be 0.03–0.30 EU/mg for rsNTX, 0.35–3.50 EU/mg for rLNTX, and 3–30 EU/mg for rCTXA3 by a Limulus amebocyte lysate test (Pyrotell, Assoc. of Cape Cod, Falmouth, MA, USA).

### 5.3. Expression of Recombinant TEV Protease

pRK793 plasmid was a gift from David Waugh (Addgene plasmid #8827; http://n2t.net/addgene:8827, accessed on 31 October 2015; RRID: Addgene_8827). The plasmid was transformed into BL21(DE3)-RIL for TEV expression. In brief, the transformed cells were cultured in LB medium containing 30 mg/L chloramphenicol and 100 mg/L ampicillin at 37 °C with shaking at 180 rpm. When bacteria reached the exponential growth phase, temperature was decreased to 30 °C, and 1 mM IPTG was added. The cells were harvested six hours after the induction and disrupted by a French press. TEV protease was purified from the soluble fraction of bacterial lysate by an IMAC column. Aliquots of TEV in 50 mM Tris (pH = 8.0) containing 300 mM NaCl and 50% glycerol (*v/v*) were stored at −20 °C.

### 5.4. Gel Electrophoresis and Immunoblotting

The purity of r3FTXs as well as the composition of snake venoms were assessed by 4–12% Bis-Tris NuPAGE (Thermo Fisher Scientific, Waltham, MA, USA) followed by Coomassie blue staining. Western blotting was also used to confirm the identity of each r3FTX and venom. Briefly, the equine antivenom (FNAV, 1:10,000) was used as the primary antibody for the rsNTX and rCTX immunoblotting, whilst NPAV (1:10,000) was used as the primary antibody for the rLNTX. Horseradish peroxidase (HRP)-conjugated goat anti-horse IgG (Jackson ImmunoResearch Laboratories, 1:10,000) was used as the secondary antibody. The activity of HRP was developed with Clarity Western ECL Substrate (Bio-Rad, Hercules, CA, USA). The protein concentration was determined by bicinchoninic acid assay according to the manufacturer’s instructions (Thermo Fisher Scientific, Waltham, MA, USA).

### 5.5. Circular Dichroism Spectroscopy Analysis

The lyophilized samples were redissolved with deionized (DI) water at a concentration of 100 μg/mL and used for structural analysis. The spectra were acquired over the wavelength range from 190 to 260 nm in a 0.5 mm path length quartz cell at 20 °C by using a circular dichroism (CD) spectrometer (JASCO J-815, Hachioji-Shi, Tokyo, Japan). Ellipticity (θ) was calculated after subtracting the blank values in millidegrees (mdeg) according to the equation [47]: ellipticity, [θ], in deg × cm^2^ × dmol^−1^  =  (mdeg × mean residue weight)/(pathlength in millimeters  ×  the concentration of protein in mg/mL), where mean residue weight = (molecular weight)/(number of residues − 1). The contents of α-helices and β-strands were estimated by the Beta Structure Selection (BeStSel) web server [48].

### 5.6. Analysis of Disulfide Bond Linkages Using Mass Spectrometry

The disulfide linkages of recombinant toxins were determined according to a previously described protocol [49] with minor modifications. Briefly, 40 μg of a protein sample was dissolved in 50 mM sodium acetate (pH = 6.0) containing 0.2% (*w/v*) RapiGest SF surfactant (Waters, MA, USA). Then, the proteins were treated with 10 mM N-ethylmaleimide (NEM) at room temperature for 30 min to block free cysteines. Then, the samples were digested at 37 °C overnight at a trypsin/protein ratio of 1:50 (*w/w*). Subsequently, the protein digest was diluted with 100 mM sodium acetate (pH = 5.0), and dimethylation labeling was performed by adding 4% (*w/v*) formaldehyde-d2 and 20 mM sodium cyanoborohydride. After incubation at room temperature for 30 min, an equal volume of 1 M HCl was added to the mixture, which then was incubated at 37 °C for 30 min and centrifuged at 22,000× *g* for 5 min to remove the RapiGest SF byproducts. An ESI-Q-TOF mass spectrometer (Synapt HDMS) connected with a nanoACQUITY UPLC system (Waters, MA, USA) was used for the analysis of disulfide bond linkages in the peptides.

Liquid chromatography (LC) separation was performed on a reverse phase (RP) C18 column (75 μm × 100 mm, 1.7 μm, Waters, MA, USA) with a trap column (180 μm × 20 mm, 5 μm, Waters, MA, USA). Mobile phases A and B were distilled water containing 0.1% formic acid (FA) and acetonitrile (ACN) containing 0.1% FA (J.T. Baker, Phillipsburg, NJ, USA), respectively. The gradient setup was as follows: 1% B to 50% B for 30 min, 50% B to 65% B from 30 to 40 min, and 65% B for 5 min for a total of 60 min separation time. A full scan was set at the m/z range of 400−1600, and six channels were used for simultaneous MS/MS detection. Raw data were processed into the peak list files using Proteome Discoverer 1.3; then, disulfide peptide search was performed using RADAR 3.0 (http://www.mass-solutions.com.tw/, accessed on 1 January 2014) based on the native sequences of 3FTXs (P60770, P01391, and P60301) with an additional glycine at the N terminus. Enzymatic cleavage at the C-terminal side of lysine and arginine allowed for up to two missed cleavages. All amines were considered deuterated. The mass tolerances were set as ±0.01 Da and ±0.2 Da for a1 and total mass, respectively. The signal intensity cutoff was set at a 10% ratio, and the maximum chain number was defined as four.

### 5.7. Separation of Venom Proteins by RP-HPLC

Venom protein separation was performed by an HPLC system (Alliance 2695; Waters, Milford, MA, USA) equipped with a dual absorbance UV detector (model 2487). The venom analyte dissolved in distilled water was loaded onto an RP column (250 × 4.6 mm, 5 µm particles with 300 Å pore size; Jupiter C18, Phenomenex, Torrance, CA, USA) and eluted at a flow rate of 0.8 mL/min by a gradient composed of two mobile phases (mobile phase B: 0.1% trifluoroacetic acid (TFA); mobile phase C: 100% ACN/0.1% TFA) as follows: 2% C for 5 min, 2–10% C for 2 min, 10–16% B for 6 min, 16–28% B for 2 min, 28–65% B for 37 min, 65–80% B for 3 min, and 2% C for 10 min. The UV absorbance of the eluate was monitored at 215 and 280 nm.

### 5.8. Animal Experiments

#### 5.8.1. Animals

Institute of Cancer Research (ICR, body weight 19–22 g) mice and New Zealand White (NZW) rabbits were purchased from BioLASCO Co., Ltd. or the Animal Health Research Institute. Animals were housed in the National Health Research Institutes (NHRI) animal center in accordance with an approved protocol (NHRI-IACUC-105108A). Horses (450–600 kg, 6 years old) were housed in a farm in Taiwan in compliance with the same animal protocol.

#### 5.8.2. Determination of Venom Lethality

Venom lethality was determined according to a previously described protocol [46]. In brief, groups of ICR mice (19–22 g, *n* = 6) were (i.p.) injected with 200 μL of serially diluted venom solution. The survival rate of mice was recorded 48 h after the venom injection, and the toxicity was expressed as the LD_50_ (μg/g, toxin amount/mouse weight) corresponding to the venom dose that caused death of 50% mice during the assay; the LD_50_ values were calculated by the trimmed Spearman–Karber method.

#### 5.8.3. Immunization of Mice with Individual r3FTX and Antibody Titration Analysis of Antiserum

Immunization was performed according to a protocol described in a previous study [50]. Initially, 50 μg of each r3FTX was detoxified with glutaraldehyde (0.25% *v/v*) and then formulated in complete Freund’s adjuvant for intramuscularly (i.m.) injecting the mice (ICR, *n* = 5/toixn). Continuous boosting was performed two times with the same amount of antigens formulated in incomplete Freund’s adjuvant at 2-week intervals. The blood was collected from the tail vein two weeks after the final immunization. The acquired serum was decomplemented (30 min at 56 °C) and then serially diluted in PBS containing 1% bovine serum albumin and added to a 96-well ELISA plate (Corning, USA) coated with 0.5 μg of toxin or venom. After incubation at room temperature for two hours, the plates were washed four times with PBS containing 0.05% (*v/v*) Tween 20 (PBST) and incubated with goat anti-mouse or goat anti-horse immunoglobulin G (IgG) conjugated with horseradish peroxidase for one hour; then, the signal was developed by adding 100 μL of TMB (3,3′,5,5′-tetramethylbenzidine) as a peroxidase substrate (KPL, Gaithersburg, MD, USA). The plates were incubated for 15 min, and the reaction was stopped by adding 50 μL of sulfuric acid (2 N H_2_SO_4_); the absorbance at 450 nm (O.D. 450 nm) was measured by a microplate reader (Sunrise, Tecan). ELISA endpoint titers were obtained from a titration curve by interpolation and defined as the reciprocal of the serum dilution corresponding to an O.D. 450 nm value of 0.3. All measurements were performed in triplicate, and the results are shown as the mean ± standard deviation (S.D.).

#### 5.8.4. Immunization with r3FTXs Mixture in Rabbit

A hyperimmunization was performed on the NZW rabbits by continuously injecting (i.m.) the solution of the detoxified r3FTXs (2 mg/total amount) mixed with Freund’s adjuvant at 2-week intervals. In parallel, a mixture composed of equal amounts of *N. kaothia* and *N. atra* venoms (1 mg/each) formulated with Freund’s adjuvant was prepared as immuogen to inject another rabbit for comparison in the study. Notably, the protein composition in r3FTXs mixture was prepared with weight ratio of 1:1:3 to mimic the composition of sNTX:LNTX:CTXs in the crude venom mixture (Appendix A). The rabbit serum was collected every two weeks before each immunization for antibody titration analysis as procedures described above.

#### 5.8.5. Immunoreactivity of Rabbit Anti-r3FTXs Antibodies with Venom Components

Twenty milligrams of purified rabbit antibodies in 1 mL of binding buffer (0.2 M NaHCO_3_ and 500 mM NaCl, pH = 8.3) were slowly loaded onto a pre-equilibrated HiTrap NHS-activated HP column (1 mL, GE, Uppsala, Sweden) and incubated for 1 h at room temperature. The column was blocked with ethanolamine (0.5 M in 0.5 M NaCl, pH = 8.3) and then washed three times with sodium acetate buffer (0.1 M in 0.5 M NaCl, pH = 4.0). Then, one hundred micrograms of venom in 1 mL of phosphate buffer was injected into the affinity column with immobilized antibodies at a flow rate of 0.3 mL/min, and the column was washed with 10 mL of PBS (pH = 7.5). The retained proteins were eluted by acetic acid (10%, *v/v*) at a flow rate of 1 mL/min for 10 min. All procedures were performed at 4 °C by using a fast protein liquid chromatography (FPLC) instrument (GE Healthcare, Piscataway, NJ, USA) equipped with a fraction collector (GE Healthcare) and an ultraviolet (UV) detector. The UV adsorption wavelength was set at UV 280 nm during the sample analysis. The flow through and eluted fractions were collected, lyophilized, and stored at −20 °C for further analysis.

#### 5.8.6. Hyperimmunization of Horse with r3FTX Mixture

The horse immunization procedures were conducted following the WHO guidelines for the production, control, and regulation of snake antivenom immunoglobulins [51]. A regular blood test was done, and all values were checked to be normal before the beginning of the experiment. A hyperimmunization was performed using the increasing dosages (3.75, 3.75, 6.25, 10.00, 12.12, 13.10, and 14.90 mg) of detoxified r3FTXs mixture formulated in Freund’s adjuvant by injecting subcutaneously at the horseback. For the blood collection, the horse was physically positioned in a stable before the blood collection via the jugular vein. In brief, a 19G needle with a syringe was inserted through the skin into the jugular vein at a 45 angle. By slowly aspirating the syringe, a total of 50 mL of blood was collected each time before the immunization. For the final blood collection, a 9G needle was used, and 1.5% body weight of blood was withdrawn from the jugular vein. Accordingly, the horse serum antibodies were purified by the protein A affinity column (5 mL, MabSelect, Prism A, GE Healthcare, Chicago, USA) following the procedures in the datasheet. The purified horse antibodies were dialyzed to PBS buffer and stored at 4 °C until further applications. The horse antibody titration toward the toxins or venoms was analyzed by indirect enzyme-linked immunosorbent assay (ELISA) following a previously described procedure [46].

#### 5.8.7. In Vivo Neutralization Assay

The purified antibodies from the plasma of immunized animals were serially diluted and then mixed with equal volumes of each cobra venom with defined lethal dosage (3 × LD_50_ for *N. atra*, 5 × LD_50_ for *N. kaouthia,* and 3 × LD_50_ *N. siamensis* venoms) at 37 °C for 30 min. Following, the pre-incubated mixture was centrifuged at 4000× *g* for 10 min to remove the precipitate. Then, 400 μL of the supernatant was injected i.p. into mice, and the survival rate was recorded for 48 h. The median effective dosage (ED_50_) of IgG was calculated by the trimmed Spearman–Karber method and defined as the weight ratio of venom (mg) to antibody (g) required to provide 50% survival after the venom challenge. The effectiveness of antivenom was also expressed as neutralization potency (P) corresponding to milligrams of venom neutralized per gram of antibody [50].

#### 5.8.8. Statistical Analysis

Statistical analysis of the neutralizing potencies of antivenoms was performed by one-way analysis of variance followed by a Bonferroni multiple comparisons test. An unpaired two-tailed t-test was used to compare the differences in antibody titers. The results with *p*-values < 0.05 were considered significant. All statistical analyses were performed using GraphPad Prism (version 6.01) software.

## Figures and Tables

**Figure 1 toxins-13-00556-f001:**
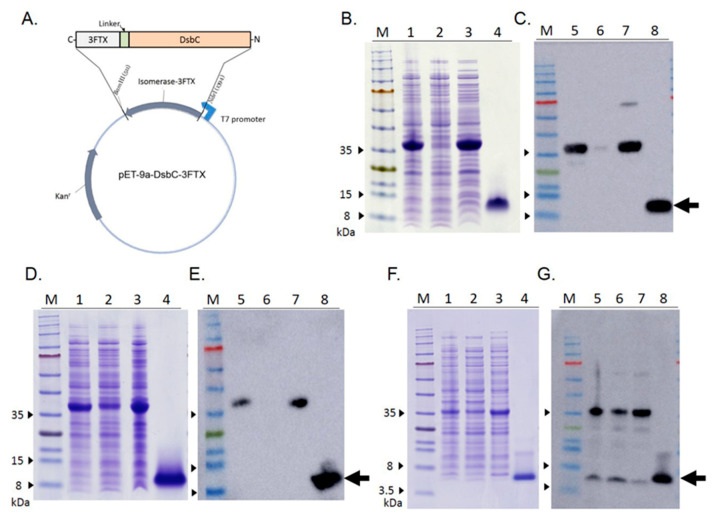
(**A**) The representative plasmid for the expression of r3FTX fused with DsbC at the N-terminus. The purity of the acquired rsNTX (**B**,**C**), rLNTX (**D**,**E**), and rCTXA3 (**F**,**G**) was confirmed by SDS-PAGE; the gels were stained with Coomassie blue, and the proteins were assayed by Western blotting with antivenom antibodies. Lane M: prestained protein markers; lanes 1 and 5: induced total lysate (total lysate after induction); lanes 2 and 6: total lysate before the induction; lanes 3 and 7: soluble fraction after cell disruption; lanes 4 and 8: purified recombinant 3FTX (indicated by an arrow).

**Figure 2 toxins-13-00556-f002:**
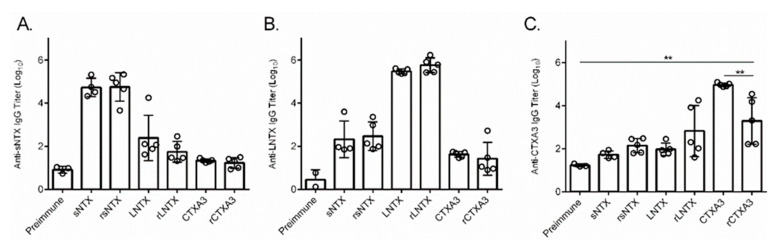
A serial dilution of the mice antiserum raised by individual 3FTX (LNTX, sNTX, and CTX A3) or individual r3FTX (rLNTX, rsNTX, and rCTXA3) was prepared to measure the antibody titers against coating antigen of (**A**) sNTX, (**B**) LNTX, or (**C**) CTXA3. Open circles indicate the titers for each mouse, and pre-immune mice sera was taken as blank in the assay. Error bars correspond to S.D. in each group. ** *p* < 0.01.

**Figure 3 toxins-13-00556-f003:**
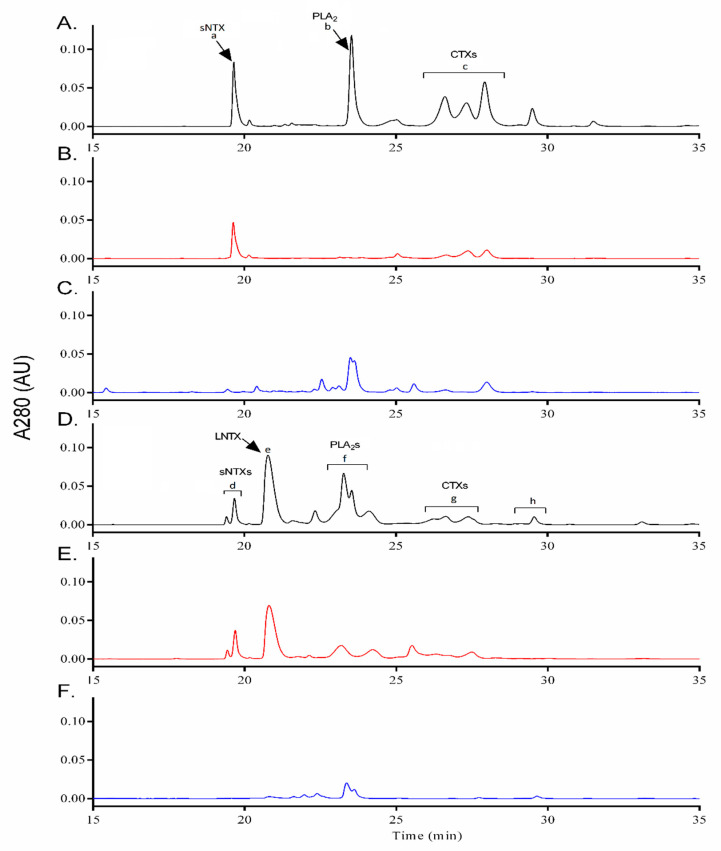
Antivenomic analysis of rabbit anti-r3FTXs antibodies HPLC chromatograms (**A**,**D**) present the protein profile of *N. atra* and *N. kaouthia* crude venoms, respectively. The protein identity of the main chromatographic peak fractions (indicated as a to h) was based on the results of proteomic characterization of the present study (Appendix A). Chromatograms (**B**,**E**) show the profiles of the immunocaptured toxins in the fractions eluted from the affinity column loaded with *N. atra* and *N. kaouthia* venoms (100 μg/each), respectively. Chromatograms (**C**,**F**) present the protein profiles of the flow through fractions from the affinity column loaded with *N. atra* and *N. kaouthia* venoms, respectively.

**Figure 4 toxins-13-00556-f004:**
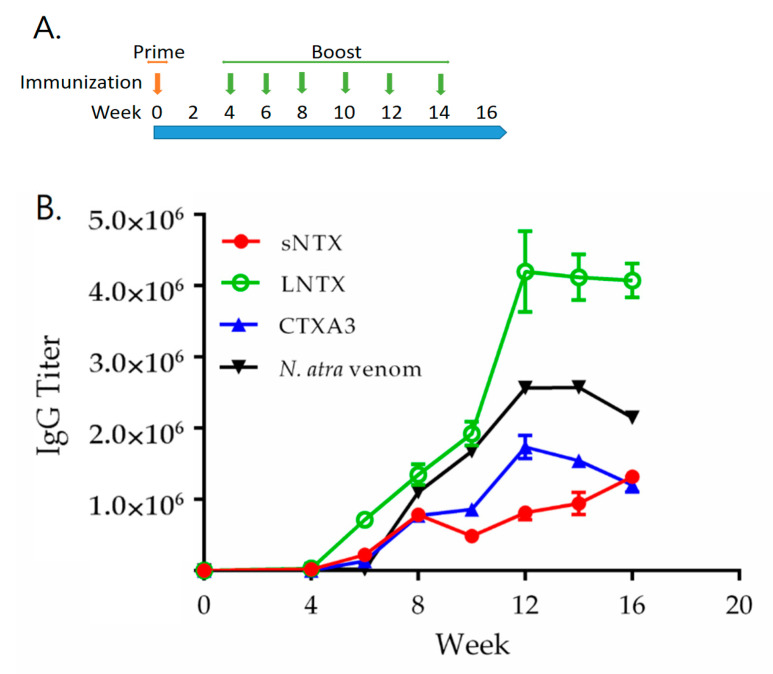
Horse immunization strategy and horse antiserum analysis. (**A**) Horse was prime immunized with r3FTXs on the first with Freund complete adjuvant via subcutaneous injection. Four weeks later after prime immunization, horse was boosted with the defined dosage of r3FTXs on the following six immunizations with Freund’ incomplete adjuvant at a 2-week interval. (**B**) The titers of specific antibody in horse serum after each immunization was examined by ELISA coated with either native LNTX, sNTX, CTX A3, or *N. atra* venom. Error bars stand for S.D. of each point. The horse serum collected at week 16 were analyzed for the neutralization titers.

**Table 1 toxins-13-00556-t001:** Structural characteristics and toxicity of r3FTXs and native 3FTXs.

		Protein			Disulfide-Linked Peptide		
Toxin	Expected MW (Da) ^1^	Observed MW (Da)	Toxicity	Secondary Structure ^3^	Peptide Sequence	Mass Ion (*m/z*)	Linkage Site(s)
LNTX	7815.66	7815.57	LD_50_ ^2^: 0.15	33.5% β-sheet	TWC^26^DAFC^30^SIR	616.2 (2+)	C26-30
			(0.12–0.19)	9.8% α-helix	C^3^FITPDITSK	846.9 (7+)	C3-C14
				10.5% Turn	DC^14^PNGHVC^20^YTK		C20-C41
					VDLGC^41^AATC^45^PTVK		C45-C56
					TGVDIQC^56^C^57^STDNC^62^NPFPTR		C57-C62
rLNTX	7872.68	7872.58	LD_50_: 0.21	34.7% β-sheet	TWC^27^DAFC^31^SIR	616.2 (2+)	C27-31
			(0.18–0.24)	9.2% α-helix	C^4^FITPDITSK	846.9 (7+)	C4-C15
				9.9% Turn	DC^15^PNGHVC^21^YTK		C21-C42
					VDLGC^42^AATC^46^PTVK		C46-C57
					TGVDIQC^57^C^58^STDNC^63^NPFPTR		C58-C63
sNTX	6945.04	6944.97	LD_50_: 0.23	54.5% β-sheet	LEC^3^HNQQSSQTPTTTGC^17^SGGETNC^24^YK	899.9 (6+)	C3-C24
			(0.17–0.26)	0.0% α-helix	GC^41^GC^43^PSVK		C17-C41
				7.4% Turn	NGIEINC^54^C^55^TTDR		C43-C54
					C^60^NN		C55-C60
rsNTX	7001.06	7000.99	LD_50_: 0.25	53.4% β-sheet	GLEC^4^HNQQSSQTPTTTGC^18^SGGETNC^25^YK	909.3 (6+)	C4-C25
			(0.19–0.33)	0.0% α-helix	GC^42^GC^44^PSVK		C18-C42
				8.1% Turn	NGIEINC^55^C^56^TTDR		C44-C55
					C^61^NN		C56-C61
CTXA3	6734.43	6735.36	IC_50_ ^4^: 1.93	54.5% β-sheet	LKC^3^NK	701.9 (2+)	C3-C21
			(1.67–2.23)	0.0% α-helix	NLC^21^YK		
				7.4% Turn	TC^14^PAGK	561.4 (5+)	C14-C38
					GC^38^IDVC^42^PK		C42-C53
					YVC^53^C^54^NTDR		C54-C59
					C^59^N		
rCTXA3	6791.45	6792.38	IC_50_ ^4^: 2.02	53.4% β-sheet	GLKC^4^NK	487.3 (3+)	C4-C22
			(1.88–2.15)	0.0 % α-helix	NLC^22^YK		
				8.1% Turn	TC^15^PAGK	561.5 (5+)	C15-C39
					GC^39^IDVC^43^PK		C43-C54
					YVC^54^C^55^NTDR		C55-C60
					C^60^N		

^1^ Oxidized cysteine was considered in the calculation of the expected MW of the protein. ^2^ LD_50_ (μg/g, toxin amount/mouse body weight) indicates the amount of toxin (i.p.) that caused half of the ICR mice (19–22 g body weight, *n* = 6) death, which were calculated by the trimmed Spearman–Karber method. Values in brackets represent 95% confidence intervals. ^3^ Estimation by the BeStSel web server [30]. ^4^ The toxicity of cardiotoxin was expressed as the IC_50_ value (μg/g) corresponding to the concentration of toxin that caused half of human promyeloblast cells (HL-60 cell, 1 × 10^5^/100 μL per toxin) lysis as described in a previous study [33].

**Table 2 toxins-13-00556-t002:** Antibody titrations of rabbit antiserum and neutralization potency of serum antibodies.

Immunogen ^1^/Assays	Antibody Titer ^2^			Potency					
	Anti-sNTX	Anti-LNTX	Anti-CTXA3	*Naja atra*^3^ × LD_50_		*Naja kaouthia* 5 × LD_50_		*Naja siamensis* 3 × LD_50_	
				ED_50_ ^4^ (mg)	P ^5^ (mg/g)	ED_50_ (mg)	P (mg/g)	ED_50_ (mg)	P (mg/g)
Venoms	1.2 × 10^5^	6.6 × 10^5^	1.3 × 10^5^	3.3	8.3	4.9	4.7	1.8	6.5
r3FTXs	1.9 × 10^5^	6.0 × 10^5^	2.9 × 10^5^	<2.5	>20	3.8	6.1	1.4	8.2
Preimmune	<10^3^	<10^3^	<10^3^	N.E. ^3^	N.E.	N.E.	N.E.	N.E.	N.E.

^1^ Venoms indicates the mixture of *N. atra* and *N. kaouthia* venoms. r3FTXs represents the combination of rsNTX, rLNTX, and rCTXA3. Pre-immune rabbit serum was used as a control, and its titer was defined as a nonspecific signal. ^2^ Bold numbers represent the highest antibody titer in the comparison set. ^3^ N.E. indicates not effective in neutralization of venom lethality. All experiments were performed in triplicate. ^4^ The ED_50_ indicates the effective amount (mg) of rabbit serum antibodies required to provide 50% of mice (ICR mice, 19–22 g body weight, *n* = 6/venom) survival after challenge with 3 × LD_50_ of *N. atra*, 3 × LD_50_ of *N. siamensis*, or 5 × LD_50_ of *N. kaouthia* venoms. Values in brackets represent 95% confidence intervals. ^5^ The neutralizing potency (P, mg venom/g antibodies) corresponds to the amount of venom (mg) neutralized by 1 g of antibodies.

**Table 3 toxins-13-00556-t003:** The neutralization potency of horse anti-r3FTXs antibodies against the lethality of cobra venoms.

	*N. atra* 3 × LD_50_		*N. kaouthia* 5 × LD_50_		*N. siamensis* 3 × LD_50_	
**Horse anti-r3FTXs antibodies**	**ED_50_ (mg) ^2^**	**P (mg/g)**	**ED_50_ (mg)**	**P (mg/g)**	**ED_50_ (mg)**	**P (mg/g)**
	1.26 (0.58–2.76) ^1^	21.33	3.78 (2.50–5.71) ^1^	6.10	0.45	25.78

^1^ Values in brackets represent 95% confidence intervals. ^2^ The ED50 indicates the effective amount (mg) of antibodies required to provide 50% survival of mice (ICR mouse, 19–22 g body weight g, *n* = 6/venom) challenged with 3 × LD50 of *N. atra*, 3 × LD50 *N. siamensis*, or 5 × LD50 *N. kaouthia* venoms.

## Data Availability

The authors confirm that the data supporting the findings of this study are available within the article and/or its Appendix A.

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
