# Peer review of "Development of a Broad-Spectrum Antiserum against Cobra Venoms Using Recombinant Three-Finger Toxins"

_toxins, 2021, doi:10.3390/toxins13080556_

Round 1
Reviewer 1 Report
Treatment of neurotoxic envenoming in Asia is often frustrating for several reasons. The weak effects of currently available antivenoms is one reason and looking for alternative treatment is important. Using recombinant venom components like recombinant alpha neurotoxins as immunogens to produce more potent antivenoms against the relevant venom components is one way to improve treatment for neurotoxic envenoming, although we still need to use horses for the final product.
The authors selected 3 Naja species, which are present in several southeast and east asian countries, such as Taiwan, China, Lao PDR, Vietnam, Thailand and Malaysia. Highly effective Antivenom against these three species will be appreciated.
The authors showed the effectivty of their product (antibodies against r-3FTXs). The important question would be whether the new product is superior over the products already available (e.g. Taiwanese Naja atra AV or Thai Naja kaouthia antivenom or Vietnamese Naja kaouthia antivenom). Are there less antibodies (IgG or Fab2) against Type I or type II alpha neurotoxins or against Cardiotoxins in these commercially available antivenoms. Is the new product more effective against the neurotoxicity of cobra venom.
I have some comments/questions:
Line 199
Why do the authors use Naja sumatrana (instead of naja siamensis) venom to test rabbit anti-r3FTXs antibodies.
Line 259
The authors stated that cross species neutralization capability of currently available commercial antivenom products is controversial. Could you clarify which products are meant and the respective references.
Line 490-496 (Methods)
Preparation of the horse immunization process seems to be crucial in the generation of an immunogenic response in horses. It has been shown that detoxification with aldehydes has an negative impact. What did the authors mean with detoxified mixture of three r-3FTXs?
What kind of Freund's adjuvant has been used for vaccination? Incomplete or Complete? The adjuvant is crucial in the vaccination process in order to enhance the production of antibodies, particularly for venom components with low molecular mass, such as alpha neurotoxins. Unfortunately complete Freund's adjuvant has inacceptable side effects when used in horses and incomplete Freund's adjuvant is less effective.
Author Response
We thank the reviewers for their helpful feedback. Please see our response to reviewer’s comments below.
Comments from the editors and reviewers:
-Reviewer #1:
Treatment of neurotoxic envenoming in Asia is often frustrating for several reasons. The weak effects of currently available antivenoms is one reason and looking for alternative treatment is important. Using recombinant venom components like recombinant alpha neurotoxins as immunogens to produce more potent antivenoms against the relevant venom components is one way to improve treatment for neurotoxic envenoming, although we still need to use horses for the final product.
The authors selected 3 Naja species, which are present in several southeast and east asian countries, such as Taiwan, China, Lao PDR, Vietnam, Thailand and Malaysia. Highly effective Antivenom against these three species will be appreciated.
The authors showed the effectivty of their product (antibodies against r-3FTXs). The important question would be whether the new product is superior over the products already available (e.g. Taiwanese Naja atra AV or Thai Naja kaouthia antivenom or Vietnamese Naja kaouthia antivenom).
Response: In the study, we demonstrated the immunization with recombinant toxins can induce the effective antibodies against three cobra venoms that composed of different 3FTXs composition. According to Leong et al study, Thai Naja kaouthia antivenom of NPAV is capable of neutralizing the N.atra (2.5 x LD50) and N. siamensis (5 x LD50) venoms with the potency of 0.52, 0.94 and 1.15 mg/ml (PLoS Negl Trop Dis. 2012;6(6):e1672),respectively, which can be re-calculated to be 6.9, 12.4 and 15.2 mg/g (mg venom/g antivenom) by the protein concentration of 75.3 mg/ml (of reconstituted antivenom, 10ml, Sci Rep. 2016,21,37299). In comparison, the performance of our horse antiserum is more effective to against N. atra and N. siamensis venoms. For Taiwanese Naja atra antivenom, Liu et al. demonstrated it is able to cross neutralize the N. kaouthia (1MLD, s.c.) and N. Siamensis (1MLD, s.c.) with the potency of 2.19 and 0.85 (ER50, mg/ml, the amount of antivenom that gives 50% survival of mice under challenge with 1MLD venom) which can be re-calculated to be 58.4 and 22.6 mg/g (mg venom/g antivenom) by the concentration of 37.5 mg/ml (of reconstituted antivenom, 10ml, quantified by BCA). Further, we examined the cross-neutralization capabilities of Taiwanese Naja atra antivenom to against high dose of N. Kaouthia venom (5 x LD50, i.p.) and showed the potency was reduced to the value of 2.4mg/g. Overall, the horse antiserum raised by r3FTX has a comparable or better potency as compared to the cross neutralization efficacy of current products against cobra venoms.
- Are there less antibodies (IgG or Fab2) against Type I or type II alpha neurotoxins or against Cardiotoxins in these commercially available antivenoms. Is the new product more effective against the neurotoxicity of cobra venom.
Response: Basically, the abundance of toxin-specific antibodies in antivenom is highly correlated with the content of 3FTXs in the immunogen. For example, Wong et al. compared the cross neutralization efficacy of three cobra antivenoms to against the Pakistani spectacled cobra which contains (long neurotoxins [LNTXs], 28.3%; short neurotoxins [SNTXs], 8%), cytotoxins (CTXs) (31.2%), and acidic phospholipases A2 (12.3%). Of them, the Taiwanese product was the least effective although N. atra (Taiwan) venom is known to contain abundant of SNTXs (Am J Trop Med Hyg., 2016, 94, 1392). Thus, the investigation demonstrate that antivenom produced against predominantly SNTXs would have limitation in neutralizing venom with abundant LNTXs. A similar phenomenon was found in the Thai product that was led effective to neutralize the sea snake venom with abundant of SNTX (J Proteomics. 2015,3;126). Therefore, optimize the immunogen with suite amount of principle 3FTXs seems to be critical to develop the antivenom with broad spectrum potency.
Neurotoxicity is commonly found in victims envenoming by cobra species with abundant of SNTX or LNTX which belong to the type of post-synaptic neurotoxins that can inhibit the interaction of ACh with the skeletal muscle nicotinic receptor. In the study, we observed that mice injected with cobra venoms would dead suddenly, which is a typical syndrome of neurotoxicity. In the neutralization assay, mice injected with cobra venoms preincubated with the horse anti-rsFTX antibodies were survived, which suggested that antibodies raised by r3FTXs could block the interaction of NTXs with the Ach receptors. However, it is hardly to know whether the new product possess the better potency to neutralize the neurotoxicity. In the future, these assays will be conducted to assess the potency of crude venom immunogens for potency comparison.
I have some comments/questions:
3. Line 199. Why do the authors use Naja sumatrana (instead of naja siamensis) venom to test rabbit anti-r3FTXs antibodies.
Response: Thank you for the indication. The typo is corrected with the venom of N. siamensis.
Line 259. The authors stated that cross species neutralization capability of currently available commercial antivenom products is controversial. Could you clarify which products are meant and the respective references.
Response: Thank you for the indication. To avoid the misunderstanding, the sentence was rephrased as follow:
“the cross-species neutralization capability of currently available commercial antivenom products is generally effective to the venom the similar toxin composition as immunogen” (line265, page 8).
- Line 490-496. Preparation of the horse immunization process seems to be crucial in the generation of an immunogenic response in horses. It has been shown that detoxification with aldehydes has an negative impact. What did the authors mean with detoxified mixture of three r-3FTXs?
Response: Thank you for the comments. In general, the venom toxins can cause local or systemic toxicity when injected to animals at the immunization course. Following Taiwan CDC guideline, glutaraldehyde was used to inactivate the toxicity of cobra venoms and recombinant toxins used for animal immunization. In the study, a low amount of glutaraldehyde (0.25%, v/v) was used to inactivate the toxicity of 3FTXs, which might reduce the possible side effect on animals.
The detoxified mixture of three r-3FTXs stands for the immunogen for horse immunization which was a mixture three recombinant proteins (rSNTX, rLNTX, and rCTXA3) detoxified by glutaradehyde in advance. For clarification, the sentence was rephrased with the detoxified r3FTXs mixture (line 503, page 13) in the article.
What kind of Freund's adjuvant has been used for vaccination? Incomplete or Complete? The adjuvant is crucial in the vaccination process in order to enhance the production of antibodies, particularly for venom components with low molecular mass, such as alpha neurotoxins. Unfortunately complete Freund's adjuvant has inacceptable side effects when used in horses and incomplete Freund's adjuvant is less effective.
Response: In the animal immunization experiments, complete Freund's adjuvant (CFA) is only used in the prime and incomplete Freund's adjuvant (IFA) was used in the following boost immunizations. For horse hyperimmunization, the Freund’d adjuvant was delivered s.c. to mitigate the potential for muscle damage. To ensure the welfare of animals, all procedures were supervised by veterinarians in the facility and mice were inspected at least two times per day by investigators.
Reviewer 2 Report
Development of a broad-spectrum antiserum against cobra venoms using recombinant three-finger toxins
Referee comments
In this study, the authors used the recombinant types of 3FTx to generate a new immunogen formulation and investigated the potency of the resulting antiserum to against the venom lethality of three medially important cobras in Asia. The authors present an interesting methodology to conduce the folding of r3FTXs in a correct way by fusion the protein 13 and the low temperature. Moreover, this study demonstrated that the horse and rabbit antiserum raised by the r3FTX mixture are able to neutralize the venom lethality of the selected three medically important cobras. The data are very interesting and this manuscript is suitable for publication. The results support the conclusions. However, in my point of view, the authors should deepen in the discussion the biochemical and biophysical issues of the success reached in the correct way for folding r3FTx discussing with the literature the difficulties of this process. Congratulations to the authors and I'm sending you some notes that can improve the manuscript.
- Introduction
Lines30-33: ...approximately 12-34% of snakebite envenoming were estimated...
The reference 4 (about proteomic characterization venom) isn't coherent whit this sentence. This question should be revised in all manuscript, like reference 18 (lines 47-50).....
2- Results
Table 1
Line 142: The acronym ICR should be described in this subtitle or in Materials and Methods for easier reading
2.3. Assessment of the immunogenicity of individual r3FTX in a mouse model
Line 151: the time of whole period of immunization should be clearly present in this topic. Another question is regard about the time of two weeks of immunization is enough to raise sufficient antibodies against 3FTx. If possible, more references should be present to support this data.
2.4. Assessment of immunogenicity of r3FTX mixture in rabbit model
The period of immunization should be present in this topic
Table 2: the data about ED50%
The data on the ED50 of N. astra must be presented punctually or within a defined period (>X<Y)
2.5. Interactivity and neutralizing potency of rabbit anti-r3FTXs antibodies
Line 203-205: This statement is not clear. Are the data being compared with antibodies raised in which? horses? please make the information clear.
2.6. Hyperimmunization with r3FTX in horse model and potency evaluation of horse antiserum
Line 238: change 3rFTXs for r3FTx
Line 243: change Humeral for Humoral
- Discussion
Line: 284
duplicated to induce the induce
Line 291: ...plateau at stage after sixth r3FTX immunization ... This statement are causing some confusion. The results showed the data plotted in weeks and the authors discuss this question using the number of immunizations. I suggest the authors standardize the data for presentation and discussion.
Author Response
We thank the reviewers for their helpful feedback. Please see our response to reviewer’s comments below.
Reviewer 2.
- Introduction
Lines30-33: ...approximately 12-34% of snakebite envenoming were estimated...
The reference 4 (about proteomic characterization venom) isn't coherent whit this sentence. This question should be revised in all manuscript, like reference 18 (lines 47-50).....
Response: Thank you for the indications. The reference 4 states ~17% of bites were caused by N. atra in China. As for the reference 18, it is changed to another paper as follow:
Nirthanan S, Gwee MC. Three-finger alpha-neurotoxins and the nicotinic acetylcholine receptor, forty years on. J Pharmacol Sci. 2004 Jan;94(1):1-17. doi: 10.1254/jphs.94.1. PMID: 14745112.
2- Results
Table 1
Line 142: The acronym ICR should be described in this subtitle or in Materials and Methods for easier reading
Response: ICR is the species of mouse which is the short of Institute of Cancer Research. The acronym ICR was added in Materials and Methods.
2.3. Assessment of the immunogenicity of individual r3FTX in a mouse model
Line 151: the time of whole period of immunization should be clearly present in this topic. Another question is regard about the time of two weeks of immunization is enough to raise sufficient antibodies against 3FTx. If possible, more references should be present to support this data.
Response: The content is modified as per your suggestion. The immunization of 3FTXs at a 2-week interval is a commonly used vaccination regimens widely used in animal like horse or mouse (see reference de la Rosa G et. al., and Ramos HR et. al.,). More references were added for supporting this viewpoint.
Ref should be added:
- de la Rosa G, Olvera F, Archundia IG, Lomonte B, Alagón A, Corzo G. Horse immunization with short-chain consensus α-neurotoxin generates antibodies against broad spectrum of elapid venomous species. Nat Commun. 2019 Aug 13;10(1):3642. doi: 10.1038/s41467-019-11639-2. PMID: 31409779; PMCID: PMC6692343.
- Ramos HR, Junqueira-de-Azevedo Ide L, Novo JB, Castro K, Duarte CG, Machado-de-Ávila RA, Chavez-Olortegui C, Ho PL. A Heterologous Multiepitope DNA Prime/Recombinant Protein Boost Immunisation Strategy for the Development of an Antiserum against Micrurus corallinus (Coral Snake) Venom. PLoS Negl Trop Dis. 2016 Mar 3;10(3):e0004484. doi: 10.1371/journal.pntd.0004484. PMID: 26938217; PMCID: PMC4777291.
2.4. Assessment of immunogenicity of r3FTX mixture in rabbit model
The period of immunization should be present in this topic
Response: The content was modified as per your suggestion.
Table 2: the data about ED50%
The data on the ED50 of N. astra must be presented punctually or within a defined period (>X<Y)
Response: Thank you for the suggestion. The potency of antisera (ED50) was re-calculated by the trimmed Spearman-Karber method and presented with or without 95% confidence intervals in Table 2
2.5. Interactivity and neutralizing potency of rabbit anti-r3FTXs antibodies
Line 203-205: This statement is not clear. Are the data being compared with antibodies raised in which? horses? please make the information clear.
Response: Thank you for the indications and the content is rewritten. Based on the neutralizing potencies presented in Table 2, we would like to demonstrated that rabbit antibodies raised by the recombinant 3FTXs were capable of neutralizing the lethality of venoms, with similar or better neutralizing potencies compared to antibodies raised by the venom mixture.
2.6. Hyperimmunization with r3FTX in horse model and potency evaluation of horse antiserum
Line 238: change 3rFTXs for r3FTx
Line 243: change Humeral for Humoral
Line: 284
duplicated to induce the induce
Response: these typos are corrected. Thank you for the indications.
Line 291: ...plateau at stage after sixth r3FTX immunization ... This statement are causing some confusion. The results showed the data plotted in weeks and the authors discuss this question using the number of immunizations. I suggest the authors standardize the data for presentation and discussion.
Response: Thank you for the suggestion and Figure 4 was re-plotted with the addition of the number of immunization.